# Linkage of maternity hospital episode statistics birth records to birth registration and notification records for births in England 2005–2006: quality assurance of linkage

Victoria Coathup ,[1] Alison Macfarlane,[2] Maria Quigley[1]

¹National Perinatal Epidemiology Unit, Nuffield Department of Population Health, University of Oxford, Oxford, UK
²Centre for Maternal and Child Health Research, School of Health Sciences, City University, London, UK

**Correspondence to**
Dr Victoria Coathup;
victoria.coathup@npeu.ox.ac.uk

## ABSTRACT

**Objectives** The objectives of this study were to describe the methods used to assess the quality of linkage between records of babies' birth registration and hospital birth records, and to evaluate the potential bias that may be introduced because of these methods.

**Design/setting** Data from the civil registration and the notification of births previously linked by the Office for National Statistics (ONS) had been further linked to birth records from the Hospital Episode Statistics (HES) for babies born in England. We developed a deterministic, six-stage algorithm to assess the quality of this linkage.

**Participants** All 1 170 790 live, singleton births, occurring in National Health Service hospitals in England between 1 January 2005 and 31 December 2006.

**Primary outcome measure** The primary outcome was the number of successful links between ONS birth records and HES birth records. Rates of successful linkage were calculated for the cohort and the characteristics associated with unsuccessful linkage were identified.

**Results** Approximately 92% (1 074 572) of the birth registration records were successfully linked with a HES birth record. Data quality and completeness were somewhat poorer in HES birth records compared with linked birth registration and birth notification records. The quality assurance algorithms identified 1456 incorrect linkages (<1%). Compared with the linked dataset, birth records were more likely to be unlinked if babies were of white ethnic origin; born to unmarried mothers; born in East England, London, North West England or the West Midlands; or born in March.

**Conclusions** It is possible to link administrative datasets to create large cohorts, allowing researchers to explore important questions about exposures and childhood outcomes. Missing data, coding errors and inconsistencies mean it is important that the quality of linkage is assessed prior to analysis.

## INTRODUCTION

The use of routinely collected datasets within research has increased rapidly over the past decade as an alternative to conducting large, observational studies, which can be extremely costly and often suffer from poor recruitment and retention rates.[1] While there are many advantages to using linked administrative data for medical research, they also present challenges. One is the quality of data recorded and, in consequence, the quality of data linkage, as this is highly reliant on the availability and accuracy of personal identifiers[2] and other supporting information.

The Digital Economy Act 2017[3] was introduced with the aim of facilitating data sharing for research purposes, but only if the data have been 'deidentified' and the research is deemed to be in the public interest.[4] Without access to personal identifiers, successful linkage between datasets becomes more

challenging. Therefore, the 'Trusted Third Party' model, whereby the full identifiers are transferred to an organisation, which will link them with its own data and return the linked data to the data controller, is now the preferred method of linkage in most research projects. While Trusted Third Parties typically publish their linkage algorithms, they usually do not publish results of quality assurance (QA) of their methods. Therefore, it is essential that researchers assess the quality of linkage and validity of data prior to conducting statistical analyses.

## Strengths and limitations of this study

► This is the first study to evaluate the linkage of birth registration, birth notification and Hospital Episode Statistics (HES) birth records in England.

► We were able to build on existing work relating to the quality assurance of birth registration, birth notification and HES delivery records.

► Access to personal identifiers meant that we were able to evaluate the quality of linkage, identify poor quality matches and assess the level of bias introduced by linking these datasets.

► Accessing personal identifiers also led to delays because of the length of approval processes and the need to travel to access these data in designated secure settings. However, the Office for National Statistics is developing a remote access system for the secure setting, so future research studies may not be affected by the need to travel.

► We examined only singleton births in National Health Service hospitals so we did not assess whether the linkage of multiple births is of comparable quality. However, previous work linking birth registration records with mothers' delivery records in multiples suggests that the linkage rate is similar to that for singletons.

The work described in this paper was conducted as part of the (Tracking the Impact of Gestational Age on Health, Educational and Economic outcomes: a Longitudinal Records Linkage Study) TIGAR study, which is a population-based, record-linkage study of births and hospital admission data in England. The study aimed to estimate the association between gestational age at birth and rates of hospital admission throughout childhood. This used previously linked data from two sources.[5] The first of these was civil registration of births, a legal process in which parents register the birth and provide mainly demographic information to a specially trained local registrar of births, deaths and marriages who records the information, issues birth certificates and forwards the data to national systems. The second is the data recorded at the notification of the birth within 36 hours to the National Health Service (NHS) by the attending midwife or other birth attendant, and the baby's NHS Number, a unique identifier, is allocated. These combined Office for National Statistics (ONS) birth records were linked with birth records from Hospital Episode Statistics (HES), the national hospital discharge system for England. This was done by the data owner, the Health and Social Care Information Centre, now known as NHS Digital, as part of an earlier, larger study by City, University of London.[6]

Two HES records are generated when a baby is born in England; one for the mother and one for each baby. Each consists of an Admitted Patient Care (APC) record common to all hospital in-patient stays plus a 'tail' with information about the birth (online supplemental figure S1). The mother's HES delivery record contains APC information relating to the mother's delivery and a 'maternity' tail'. The baby's HES birth record contains the baby's APC record and a 'baby tail' containing details of the birth and overlapping extensively with the 'maternity tail'. Maternity HES data are downloaded from hospitals' administrative systems. As these do not all come from the same supplier, there are some differences in the ways in which data are entered and there are differences between systems and hospitals in the extent to which data items are missing.

The team at City, University of London, has already evaluated the quality of linkage between birth registration, birth notification and the mother's HES delivery records for births from 2005 to 2014. The authors reported a linkage rate of 95% and uncovered some linkage errors.[7] Therefore, the main objectives of the current study were to assess the quality of linkage between baby's birth registration and notification records and the baby's HES birth records, and to evaluate the potential bias introduced to the study cohort by the linkage. This has relevance for analyses of similar linked administrative datasets.

## METHODS

All live, singleton babies born in NHS hospitals between 1 January 2005 and 31 December 2006 to a mother living in England were eligible for inclusion in this study cohort. The analysis was restricted to births in NHS hospitals, but they accounted for 96.6% of women giving birth in 2006. Home births accounted for 2.7% of deliveries in 2006, but although most received NHS care, many were not recorded on hospital systems and so not included in HES. There were extremely few HES records for the 0.5% of deliveries in non-NHS hospitals and the 0.2% delivering elsewhere.[8]

All analyses were conducted using STATA V.14[9] within the ONS' Secure Research Service (SRS). An overview of the procedures involved in the QA process for this study is presented in figure 1.

### Data sources

Two datasets containing data about births in England from 1 January 2005 to 31 December 2006 were linked (Figure S1). They and the linkage file were saved as three separate STATA datasets. The datasets included: (1) ONS births; (2) HES birth records; and (3) the linkage file containing unique ONS birth identifiers (ONSID) and corresponding unique HES identifiers (HESID). These files are described in the online supplemental information (Online supplemental figure S1).

### ONS births

The master dataset comprises data from two sources: birth registration and birth notification. These two datasets have been routinely linked by the ONS since 2006 and the combined dataset is referred to as 'ONS births' throughout this paper. ONS births contains personal identifiers, sociodemographic characteristics and birth characteristics.[10]

### Hospital Episode Statistics (HES birth records)

HES is a large database containing records of all episodes of care and births in NHS hospitals in England since 1989. The records used here are from the HES APC dataset, which contains records of all inpatient admissions, including birth and delivery records, to NHS hospitals across England. A full description of the database can be found elsewhere.[11] Briefly, HES inpatient admissions are structured as 'episodes' of care, with an episode defined as a period of care under one consultant or midwife. Each episode contains details relating to the individual, care provider and care received (including diagnosis and procedural codes). If a patient receives care in more than one department, this generates multiple

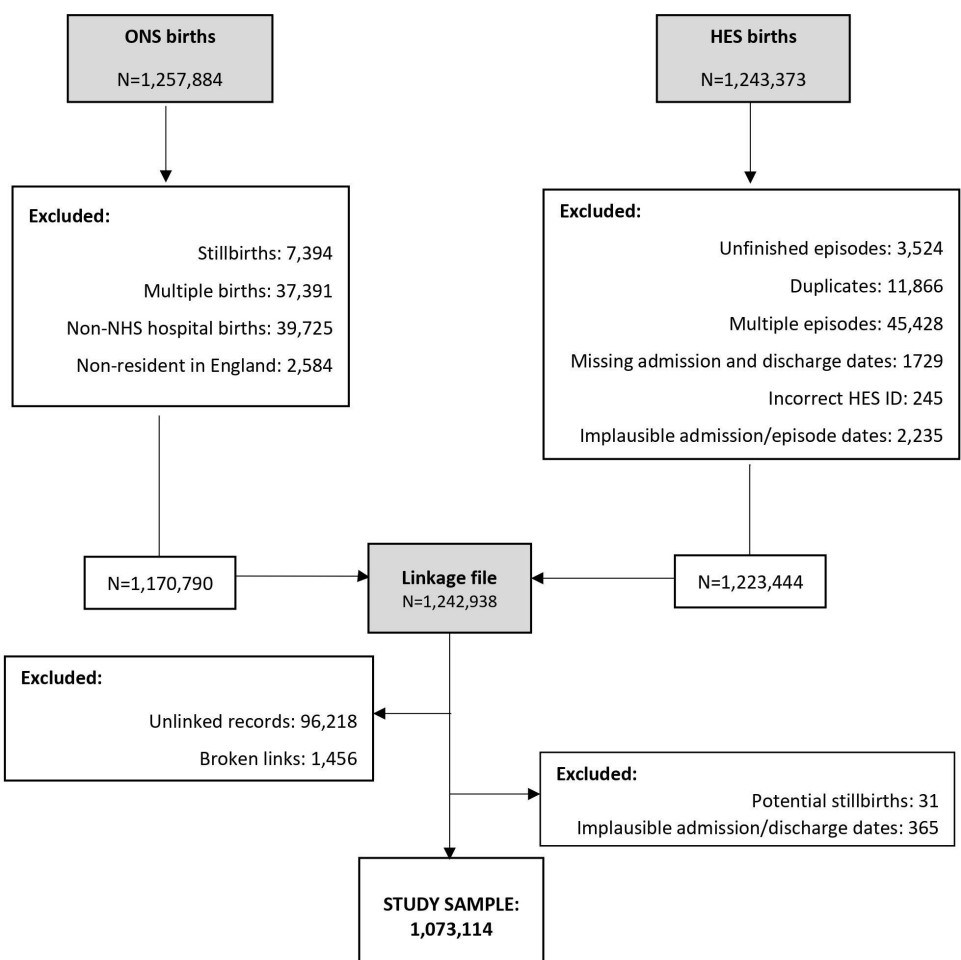

**Figure 1** Flow chart of study population. HES, Hospital Episode Statistics; ID, identifier; NHS, National Health Service; ONS, Office for National Statistics

episodes, referred to as a 'spell' and represents an uninterrupted period of care within one hospital. A new spell is generated when the patient is transferred to a different hospital to continue care. A continuous inpatient stay may consist of one or more episodes and spells, and ends when the patient is discharged from an NHS hospital. Data about hospital episodes are primarily collected for financial reimbursement, and therefore, the datasets are divided into financial years, beginning 1 April and ending 31 March. Episodes are labelled as 'finished' once the patient is discharged from hospital. However, if an episode begins in one financial year and ends during the next, two episodes will be generated; one in the financial year the episode begins and one in the financial year that the episode ends. In this case, the first episode will be defined as 'unfinished'.

Everyone for whom records are stored in HES is assigned a unique identifier, called the HESID.[12] This is generated using a combination of NHS Number, local patient identifier, postcode, sex and date of birth to enable data users to uniquely track patients throughout the NHS. Descriptions of the variables available are in NHS Digital's HES Data Dictionary.[13]

When a baby is born, the general inpatient episode becomes part of the Maternity HES dataset and 19 additional variables relating to the delivery or birth are appended. For each birth, two maternity HES records are generated. First, a HES delivery record, which includes the general inpatient record for the mother and 19 additional variables, referred to as the maternity tail. Second, a HES birth record, which includes the general inpatient record for the baby, along with 19 additional variables, referred to as the baby tail. The maternity and baby tails contain similar information relating to the delivery and birth; however, this study evaluated the linkage of HES birth records only. These additional data items in the baby tail include variables such as gestational age at birth and neonatal level of care. For a full list, see section A in the online supplemental information.

### Linkage file

The linkage file contained the unique identifiers from each dataset as linked by NHS Digital. The dataset contained unique identifiers from ONS births (ONSID) that had been successfully linked with the corresponding unique identifier from HES births (HESID) and those which had not. It also contained unique HES birth

identifiers that were not linked to a unique ONS birth identifier.

### Linkage of ONS births and Maternity HES

Linkage between ONS birth records and HES birth records is a two-step process. First, ONS birth records were linked to the HES patient index (an index of personal details relating to all individuals with access to NHS hospitals) by NHS Digital using a deterministic algorithm in order to assign the HESID; and second, records were linked to the corresponding HES birth records using the HESID and other identifiers.[5] Here, each linked record was assigned a match rank score, indicating the stage in the algorithm at which the records had been matched (one being highest quality). Steps of the algorithm are summarised in online supplemental table S1.

### Data preparation for QA checks

Full details of available variables, data cleaning procedures and preparation steps for the included datasets are described elsewhere.[6] Many key variables were available from both birth registration and birth notification. The preferred sources are summarised in the online supplemental table S2. A number of additional steps were taken to ensure that key variables in HES births and ONS births were in consistent formats, ready for comparison during the QA procedure (online supplemental table S3). Further checks were conducted on HES birth records and those with: (1) discharge date occurring before admission; (2) discharge occurring before baby's date of birth; or (3) admission dates occurring before 1 January 2005 or after 31 December 2006 were excluded.

### Multiple episodes with the same HESID (i.e. duplicate HES birth records)

Birth episodes with the same HESID can occur due to a number of reasons: (1) unfinished episodes, such as a baby being born on the 30 March, but then discharged on the 1 April[11]; (2) administrative errors[7]; (3) a birth spell containing multiple episodes, such as a baby being transferred to a different hospital or consultant; and (4)

HESID is incorrectly assigned to more than one baby.[14] The steps which were taken have been summarised in the online supplemental information. In addition to this, episode start dates that occurred more than two days after the previous episode ended and did not include a transfer code (see online supplemental information) were considered a new admission and were saved separately.

### QA of ONS births and Maternity HES linkage

After merging the three datasets, we identified cases where two different ONS birth records had been linked to the same HES birth record and key characteristics were compared to identify the correct link. This was done by comparing the following variables which were available in both data sources: baby's date of birth, gestational age, birth weight, sex, mother's date of birth and postcode. The record with the highest number of matching variables was identified as the correct link. When records matched on the same number of variables, the record that matched exactly on birth weight was identified as the correct match. If birth weight was missing or did not match with either record, then the record with the highest match rank score from the original linkage (online supplemental table S1) was identified as the correct link. The remaining records were then compared manually, but in cases where records had a high proportion of missing data or the same number of matching variables and the same match rank score, it was not possible to identify the correct link and therefore all were excluded.

The deterministic algorithm developed for evaluating the record linkage is summarised in table 1. The QA algorithm was adapted from a previous study which assessed the quality of linkage between ONS birth records and the mother's HES delivery records,[7] and was based on location of birth, baby's date of birth, sex, birth weight and gestational age, mother's date of birth and postcode. The location of birth was defined as the NHS hospital trust running the hospital that the baby was born in. An NHS hospital trust in England is an organisational unit within the NHS and usually refers to a group of hospitals that

**Table 1** Algorithm for quality assurance of linkage of Office for National Statistics birth records to Hospital Episode Statistics birth records

| **Variable** | | | | | | | | | | | | |
|---|---|---|---|---|---|---|---|---|---|---|---|---|
| **Step** | **Hospital trust code** | | **Date of birth** | | **Sex** | | **Birth weight** | | **Gestational age** | | **Mother's date of birth** | | **Postcode** |
| 1 | E | *and* | E | *and* | E | *and* | E | *and* | E | | – | | – |
| 2 | E | *and* | E | *and* | E | *and* | (E | *or* | E) | | – | | – |
| 3 | E | *and* | E | *and* | E | | – | | – | *and* | E/M | *and* | E/M |
| 4 | – | | E | *and* | E | *and* | E/M | *and* | E/M | *and* | E/M | *and* | E/M |
| 5 | E | *and* | P | *and* | E | *and* | (E | *or* | E | *or* | E | *or* | E) |
| 6 | E | *and* | P | | – | | E/M | *and* | E/M | *and* | E/M | *and* | E/M |

E, exact match; M, missing; P, partial match (differs by up to 4 days, two elements of data match or dates match if day and month swapped).

are in close proximity to each other. This was used instead of the hospital of birth to account for potential transfers between hospitals within the birth admission. The hospital trust variable was developed as part of a previous study.[6] To account for differences in rounding between hospitals, a birth weight of + or − 100 g was considered a match.

Final checks after the QA procedures were completed included: (1) checking that the baby's date of birth in ONS births was within the admission and discharge dates from the HES birth record; (2) baby's date of birth in ONS births did not occur before 1 January 2005 or after 31 December 2006; (3) hospital discharge date did not occur before the admission date; and (4) further checks to ensure that all stillbirths had been excluded (International Classification of Diseases, version 10 (ICD10) diagnosis code=Z37.1).

### Assessment of linkage bias

The distributions of key characteristics were compared between the linked and unlinked study samples and the eligible study sample. The $\chi^2$ test was used to assess whether distributions differed significantly between the linked and unlinked study samples. Because the sample sizes were so large, we also looked at differences in distributions of variables, for example, differences between proportions of more than 1%. In addition to this, we compared the distributions of key characteristics between the linked study sample and all live births in England in 2005 and 2006 to see what effect excluding stillbirths, multiple and non-NHS births had on the study cohort.

### Patient and public involvement

The TIGAR study was supported by a patient, parent and public advisory group, which provided input to different aspects of the study. This group met at the start of the study and gave input into the study protocol and the lay summary of the project.

## RESULTS

There were 1 257 884 ONS birth records. After excluding multiple births and babies who were not eligible for the TIGAR study, 1 170 790 live, singleton babies born from 1 January 2005 to 31 December 2006 in NHS hospitals to women living in England remained in the study cohort. There were 1 243 373 HES birth records and the linkage file included 1 242 938 records (figure 1).

### Linkage of ONS births to Maternity HES

Of the 1 170 790 eligible ONS birth records, 1 074 571 (92%) were successfully linked with a HES birth record. Of the 96 219 unlinked records, 88 471 had a HESID but no corresponding HES birth record and 7747 had no corresponding HESID in the linkage file. The majority of records linked had a match rank score of one, which meant they exactly matched on all four variables in the NHS Digital algorithm, with fewer than 1% of records

linking in stages 3–6 of the algorithm (online supplemental table S1). A higher proportion of links had a match rank score of one in 2006 compared with 2005, however.

### Data preparation for QA checks

All key variables in ONS birth records had data missing for less than 1% of births. In contrast, data were missing from substantial numbers of HES birth records (online supplemental table S4)

### Multiple birth episodes with the same HESID

Of the 1 243 373 HES birth records, 73 307 (6%) had linked with more than one record with the same HESID. Of these, the most common reasons for this were multiple episodes within a hospital spell (62%), unfinished episodes (15%) and duplicate episodes due to administrative errors (11%; figure 1). There were 1 197 999 unique HES birth records remaining.

### QA of the ONS births and Maternity HES linkage

A total of 979 (0.1%) HES birth records had linked with more than one ONS birth record. The high proportion of missing data in HES birth records meant that it was difficult in many cases to identify the correct match. Of these, 499 records were judged to be incorrectly linked and were broken. It was not possible to ascertain the status of 95 records, which had the same matching variables and match rank score. A lot of data items in these records were missing data and all were excluded (table 2).

Key variables of the 1 074 571 (92%) ONS birth records which had been successfully linked were then compared using the algorithm in table 1. The vast majority (99.5%) of correct links were found to be within the first three stages of the QA algorithm (table 3). The 645 records identified as correct links in stage 4 of the algorithm appeared to be correct matches; the hospital location code differed slightly, which may have resulted from either a transfer to a neighbouring hospital trust during the birth admission or from data entry errors. In stage five, 1973 (0.2%) records were identified as correct links. Of these, 76% had a match rank score of one, suggesting the date of birth had been entered incorrectly in the HES birth record. The majority of these matches looked like data entry errors where the month and day had inadvertently been swapped. In stage six of the algorithm, a small number of records were identified as correct links. These records had a lot of data missing, including 64% of birth weights, 70% of gestational ages, 77% of postcodes; and 67% of mother's dates of birth. Of the 242 records with partially matching dates of births, 91% had a match rank score of one or two, suggesting good quality matches but with data entry errors in the HES birth records. Interestingly, almost 62% of records in this stage exactly matched on date of birth, but differed by sex. Most of these records also exactly matched on birth weight, gestational age, postcode and mother's date of birth, suggesting they are

**Table 2** Proportions of records of births in 2005 and 2006 at each quality assurance stage for live, singleton births born in National Health Service (NHS) hospitals to mother's living in England

| Stage of quality assurance | | Total n | 2005 n | % | 2006 n | % |
|---|---|---|---|---|---|---|
| 1 | Number of births* | 1 170 790 | 575 568 | 49.2 | 595 222 | 50.8 |
| 2 | Number of ONS birth records linked to HESID | 1 148 110 | 563 963 | 49.1 | 584 147 | 50.9 |
| 3 | Number of ONS birth records not linked to HESID | 22 680 | 11 605 | 51.2 | 11 075 | 48.8 |
| 4 | Number of ONS birth records linked to HESID and HES birth record | 1 074 571 | 530 586 | 49.4 | 543 985 | 50.6 |
| 5 | Number of ONS birth records linked to HESID but not linked to HES birth record | 96 219 | 44 982 | 46.7 | 51 237 | 53.3 |
| 6 | Number of ONS birth records linked to one HES birth record before QA | 1 074 571 | 530 586 | 49.4 | 543 985 | 50.6 |
| 7 | Number of links broken between ONS birth records and HES birth records after QA | 1456 | 890 | 61.1 | 566 | 38.9 |
| 8 | Number of ONS birth records linked to one HES birth record after QA | 1 073 115 | 530 119 | 49.4 | 542 996 | 50.6 |
| – | % of total ONS birth records left linked to one HES birth record after QA | 91.7 | – | 92.1 | – | 91.3 |
| 9 | Total number of ONS birth records not linked to HES birth record after QA | 97 675 | 45 810 | 46.9 | 51 865 | 53.1 |
| – | % of total ONS birth records not linked to HES birth record after QA | 8.3 | – | 7.9 | – | 8.7 |

*Birth registration linked to birth notification records of live, singleton births that occurred in NHS hospitals to women residing in England
HES, Hospital Episode Statistics; HESID, HES identifiers; ONS, Office for National Statistics; QA, quality assurance.

correct links, again with data entry errors in the HES birth record.

Overall, 860 records with incorrect links were identified and 64% of these were in births in 2005, suggesting an improvement in data and linkage quality in 2006. When exploring these broken links in relation to their match rank score, 69% had a score of six, meaning their NHS numbers were missing. The main reasons for broken links were completely different dates of birth, as opposed to exact or partial matching, and data missing for large numbers of variables. Among these records, birth weight, gestational age, postcode and mother's date of birth was missing for approximately 92%, 91%, 85% and 91%,

respectively. There were a small number of records that appeared to be incorrectly broken due to cleaning errors, for example, postcodes which included an 'O' instead of a zero.

A number of interesting observations were made when reviewing the broken links: (1) many that were broken differed by gestational age, but in most cases by just 1 week and the babies were full term, for example, 39 and 40 weeks; and (2) more than one-third of broken links were for babies born in London. As the numbers of broken links were small, distributions of numbers of broken links cannot be presented, as they could be disclosive.

**Table 3** Proportion of birth records identified as correct links at each stage of the quality assurance process

| Step | Total Number | % | 2005 Number | % | 2006 Number | % |
|---|---|---|---|---|---|---|
| 1 | 385 570 | 35.9 | 207 106 | 39.1 | 178 464 | 32.8 |
| 2 | 301 066 | 28.0 | 134 684 | 25.4 | 166 382 | 30.6 |
| 3 | 383 189 | 35.6 | 186 203 | 35.1 | 196 986 | 36.2 |
| 4 | 645 | 0.1 | 426 | 0.1 | 219 | 0.0 |
| 5 | 1973 | 0.2 | 983 | 0.2 | 990 | 0.2 |
| 6 | 671 | 0.0 | 379 | 0.0 | 292 | 0.0 |
| Poor quality links | 860 | 0.1 | 548 | 0.1 | 312 | 0.1 |
| All | 1 074 571 | 100 | 530 586 | 100 | 543 985 | 100 |

## Assessment of linkage bias

Distributions of key variables for all live births, all eligible births and all linked and unlinked births are presented in table 4. Compared with the linked dataset, the distribution of those in the unlinked dataset differed by baby's ethnicity, mother's age, parity, Index of Multiple Deprivation score (IMD) score, sex, registration status, region and month of birth, birth weight and gestational age (p<0.05). Although absolute differences were small (<1%) for most variables, there were some noticeable discrepancies, where distributions differed by more than 1%. Unlinked records were more likely to occur if babies were of white ethnic origin; born outside marriage; born in East England, London, North West England or the West Midlands; or born in March. Compared with all eligible births in 2005 and 2006, the distributions of linked births were very similar and differed by less than 1% in all cases.

## DISCUSSION
### Key findings

Our study has shown that linkage between ONS births and HES births, based on NHS number; baby's date of birth, sex and postcode, is possible. Overall, 92% of ONS birth records for live, singleton births born in NHS hospitals to mothers living in England were successfully linked to a HES birth record. After checking the quality of linkage in two stages, 1456 records (<1%) were identified as potentially incorrect links. The main reasons for breaking links were different dates of birth and large amounts of data missing from key fields, making it difficult to determine if the link was correct. A higher proportion of links was broken for births in 2005 compared with births in 2006, suggesting an improvement in data quality between the two years. The addition of partially matching date of birth in the QA algorithm increased sensitivity and identified more correct linkages, which would have otherwise been missed from the study cohort.

Duplicate HES records and large amounts of data missing for some variables created challenges when assessing linkage quality, highlighting the importance of these procedures before beginning statistical analysis. Finally, birth records were more likely to not link if babies were of white, British ethnic origin, born outside marriage, born in East England, London, North West England or the West Midlands, or born in March.

### Strengths and limitations

Key strengths of this study included building on previous work conducted into the linkage of mothers' hospital records with babies' birth records, obtaining data from multiple sources, and the use of personal identifiers, which allowed us to evaluate the linkage more easily. However, the need to access personal identifiers also increased the problems involved in accessing the data, notably in terms of the length of approvals processes. Other limitations include the restriction of the assessment of linkage to live, singleton births in 2005 and 2006. We were unable to assess how well the algorithms designed for this study would perform with multiple births. These tend to have more complex data, although similar algorithms were successfully used for the quality of assurance of the linkage of multiple births to mothers' delivery records.[6] In addition, the algorithms used in this study were deterministic, rather than probabilistic. The latter can be more effective when dealing with complex records, such as those with a lot of missing data or coding errors.[15 16]

### Interpretation of findings

Approximately 8% of ONS birth records did not link to a HES birth record. One of the key reasons for unlinked records appeared to be missing data in key linking fields. Quality and completeness are always a concern when using routinely collected datasets and the large sample sizes they provide are always traded off with these limitations. Data quality was somewhat poorer in HES birth records than in ONS birth records and this is a commonly cited issue when working with HES data.[17 18] In studies using HES birth data in later years, the data quality improves with time. In 2009/2010, the baby tail is far more complete, with only 18% and 14% of births with missing gestational age and birth weight, respectively.[17] Therefore, researchers who wish to analyse the linked data for the remaining years (2007–2014) should find a higher linkage rate with better quality linkage. However, in previous work with mother's HES delivery records, the linkage rate plateaued at around 98% between 2010 and 2014 for singleton births, whereas the linkage rate in multiples began to decrease from 2010.[7]

Compared with the full linked dataset, records of babies born in East England, North West England, London or the West Midlands were more likely to not link. It is likely that these variations are due to differences between hospital trusts in the ways in which definitions or protocols are used, because errors occur during the transfer of data from one organisation to another, or in the overall quality of Maternity HES for certain hospitals. It is also possible that regional reporting differences account for the higher proportion of white babies and births outside marriage in the unlinked dataset. Babies born in March were also more likely to have unlinked records and this was also the case with the mothers' HES delivery records.[7] HES uses financial years (1 April–31 March) for report, so differences in reporting standards prior to the financial year-end may account for this.

We found a small proportion of records that were potentially incorrect linkages, suggesting that linkage performed by 'Trusted Third Parties', such as NHS Digital, may not be 100% accurate. Therefore, it is essential that researchers understand the quality of linkage undertaken by a Trusted Third Party prior to performing any statistical analysis. While the responsibility should fall to the Trusted Third Party to conduct quality assessment of its linkage methods and make them publicly available to researchers, this does not happen in practice.[7] The

**Table 4** Comparison of all live, all eligible, linked and unlinked datasets

| Characteristics | All live births (n=1 250 490) | | All eligible births (n=1 170 790) | | Linked (n=1 073 115) | | Unlinked (n=97 675) | | P value* |
|---|---|---|---|---|---|---|---|---|---|
| | n | % | n | % | n | % | n | % | |
| **Ethnicity** | | | | | | | | | |
| Bangladeshi | 16 730 | 1.3 | 16 281 | 1.4 | 15 356 | 1.4 | 925 | 1.0 | <0.0001 |
| Indian | 32 241 | 2.6 | 31 287 | 2.7 | 28 970 | 2.7 | 2318 | 2.4 | |
| Pakistani | 48 570 | 3.9 | 47 352 | 4.0 | 43 702 | 4.1 | 3648 | 3.7 | |
| Black African | 24 461 | 2.0 | 23 394 | 2.0 | 21 504 | 2.0 | 1888 | 1.9 | |
| Black Caribbean | 8668 | 0.7 | 8200 | 0.7 | 7420 | 0.7 | 781 | 0.8 | |
| White British | 788 985 | 63.0 | 738 887 | 63.1 | 675 340 | 62.9 | 63 552 | 65.1 | |
| White other | 91 311 | 7.3 | 86 918 | 7.4 | 80 611 | 7.5 | 6306 | 6.5 | |
| Other | 102 048 | 8.2 | 97 362 | 8.3 | 89 043 | 8.3 | 8318 | 8.5 | |
| Missing | 137 476 | 11.0 | 121 109 | 10.3 | 111 169 | 10.4 | 9939 | 10.2 | |
| **Mother's age** | | | | | | | | | |
| Under 20 | 52 605 | 4.2 | 51 312 | 4.4 | 46 803 | 4.4 | 4509 | 4.6 | <0.0001 |
| 20–24 | 216 768 | 17.3 | 208 688 | 17.8 | 190 729 | 17.8 | 17 960 | 18.4 | |
| 25–29 | 306 603 | 24.5 | 290 132 | 24.8 | 266 002 | 24.8 | 24 130 | 24.7 | |
| 30–34 | 363 312 | 29.1 | 337 323 | 28.8 | 309 604 | 28.9 | 27 718 | 28.4 | |
| 35–39 | 244 952 | 19.6 | 223 256 | 19.1 | 204 990 | 19.1 | 18 263 | 18.7 | |
| 40+ | 66 250 | 5.3 | 60 079 | 5.1 | 54 987 | 5.1 | 5095 | 5.2 | |
| **Nulliparous** | | | | | | | | | |
| No | 587 834 | 47.0 | 562 207 | 48.0 | 523 052 | 48.7 | 39 152 | 40.1 | <0.0001 |
| Yes | 570 129 | 45.5 | 548 258 | 46.8 | 505 980 | 47.2 | 42 279 | 43.3 | |
| Missing | 92 527 | 7.4 | 60 325 | 5.2 | 44 083 | 4.1 | 16 244 | 16.6 | |
| **IMD Score (quintiles)** | | | | | | | | | |
| Q1 | 333 809 | 26.7 | 318 830 | 27.2 | 292 312 | 27.2 | 26 513 | 27.1 | 0.001 |
| Q2 | 263 212 | 21.1 | 247 779 | 21.2 | 227 080 | 21.2 | 20 700 | 21.2 | |
| Q3 | 221 720 | 17.7 | 206 648 | 17.7 | 189 758 | 17.7 | 16 891 | 17.3 | |
| Q4 | 200 891 | 16.1 | 186 165 | 15.9 | 170 313 | 15.9 | 15 851 | 16.2 | |
| Q5 | 195 382 | 15.6 | 181 041 | 15.5 | 165 766 | 15.5 | 15 277 | 15.6 | |
| Missing | 35 476 | 2.8 | 30 327 | 2.6 | 27 886 | 2.6 | 2443 | 2.5 | |
| **Sex** | | | | | | | | | |
| Male | 639 617 | 51.2 | 599 715 | 51.2 | 549 232 | 51.2 | 50 484 | 51.7 | 0.002 |
| Female | 610 873 | 48.9 | 571 075 | 48.8 | 523 883 | 48.8 | 47 191 | 48.3 | |
| **Mother UK born** | | | | | | | | | |
| No | 274 659 | 22.0 | 259 171 | 22.1 | 237 663 | 22.2 | 21 504 | 22.0 | 0.413 |
| Yes | 974 065 | 77.9 | 909 961 | 77.7 | 833 943 | 77.7 | 76 022 | 77.8 | |
| Missing | 1766 | 0.1 | 1658 | 0.1 | 1509 | 0.1 | 149 | 0.2 | |
| **Registration status** | | | | | | | | | |
| Married | 716 858 | 57.3 | 665 764 | 56.9 | 612 063 | 57.0 | 53 698 | 55.0 | <0.0001 |
| Sole registration | 85 500 | 6.8 | 81 080 | 6.9 | 74 158 | 6.9 | 6922 | 7.1 | |
| Joint registration, same address | 339 641 | 27.2 | 320 098 | 27.3 | 292 338 | 27.2 | 27 762 | 28.4 | |
| Joint registration, different address | 108 491 | 8.7 | 103 848 | 8.9 | 94 556 | 8.8 | 9293 | 9.5 | |
| **Region of birth** | | | | | | | | | |
| East Midlands | 87 310 | 7.0 | 82 311 | 7.0 | 77 750 | 7.3 | 4561 | 4.7 | <0.0001 |
| East of England | 126 925 | 10.1 | 118 491 | 10.1 | 106 961 | 10.0 | 11 530 | 11.8 | |
| London | 248 341 | 19.9 | 228 980 | 19.6 | 208 222 | 19.4 | 20 756 | 21.3 | |
| North East | 61 007 | 4.9 | 57 491 | 4.9 | 54 569 | 5.1 | 2922 | 3.0 | |
| North West | 170 504 | 13.6 | 159 859 | 13.7 | 142 855 | 13.3 | 17 003 | 17.4 | |
| South Central | 88 798 | 7.1 | 83 610 | 7.1 | 78 451 | 7.3 | 5159 | 5.3 | |
| South East Coast | 96 606 | 7.7 | 90 875 | 7.8 | 83 956 | 7.8 | 6921 | 7.1 | |

Continued

**Table 4** Continued

| Characteristics | All live births (n=1 250 490) | | All eligible births (n=1 170 790) | | Linked (n=1 073 115) | | Unlinked (n=97 675) | | P value* |
|---|---|---|---|---|---|---|---|---|---|
| | n | % | n | % | n | % | n | % | |
| South West | 107 376 | 8.6 | 101 120 | 8.6 | 98 438 | 9.2 | 2682 | 2.8 | |
| West Midlands | 137 859 | 11.0 | 129 223 | 11.0 | 108 062 | 10.1 | 21 160 | 21.7 | |
| Yorkshire/Humber | 125 765 | 10.1 | 118 830 | 10.2 | 113 851 | 10.6 | 4981 | 5.1 | |
| Month of birth | | | | | | | | | |
| January | 100 604 | 8.1 | 94 233 | 8.1 | 85 746 | 8.0 | 8487 | 8.7 | <0.0001 |
| February | 93 456 | 7.5 | 87 683 | 7.5 | 78 333 | 7.3 | 9350 | 9.6 | |
| March | 104 069 | 8.3 | 97 526 | 8.3 | 85 779 | 8.0 | 11 747 | 12.0 | |
| April | 99 977 | 8.0 | 93 633 | 8.0 | 86 701 | 8.1 | 6933 | 7.1 | |
| May | 104 992 | 8.4 | 98 313 | 8.4 | 90 856 | 8.5 | 7455 | 7.6 | |
| June | 105 124 | 8.4 | 98 149 | 8.4 | 90 082 | 8.4 | 8071 | 8.3 | |
| July | 108 249 | 8.7 | 101 330 | 8.6 | 93 908 | 8.8 | 7421 | 7.6 | |
| August | 109 968 | 8.8 | 103 229 | 8.8 | 95 674 | 8.9 | 7555 | 7.7 | |
| September | 109 717 | 8.8 | 102 752 | 8.8 | 94 918 | 8.9 | 7833 | 8.0 | |
| October | 108 485 | 8.7 | 101 301 | 8.6 | 94 421 | 8.8 | 6881 | 7.0 | |
| November | 102 517 | 8.2 | 95 848 | 8.2 | 87 938 | 8.2 | 7910 | 8.1 | |
| December | 103 332 | 8.3 | 96 793 | 8.3 | 88 759 | 8.3 | 8032 | 8.2 | |
| Birth weight (g) | | | | | | | | | |
| <1500 | 15 130 | 1.2 | 11 073 | 1.0 | 10 016 | 0.9 | 1056 | 1.1 | <0.0001 |
| 1500–1999 | 19 282 | 1.5 | 13 439 | 1.2 | 12 454 | 1.2 | 985 | 1.0 | |
| 2000–2499 | 60 012 | 4.8 | 47 825 | 4.1 | 44 089 | 4.1 | 3738 | 3.8 | |
| 2500–2999 | 213 073 | 17.0 | 195 554 | 16.7 | 179 236 | 16.7 | 16 319 | 16.7 | |
| 3000–3499 | 447 030 | 35.8 | 427 600 | 36.5 | 391 926 | 36.5 | 35 675 | 36.5 | |
| 3500–3999 | 357 505 | 28.6 | 342 606 | 29.3 | 314 060 | 29.3 | 28 540 | 29.2 | |
| 4000–4499 | 116 484 | 9.3 | 111 708 | 9.5 | 102 534 | 9.6 | 9176 | 9.4 | |
| 4500+ | 20 730 | 1.7 | 19 940 | 1.7 | 18 328 | 1.7 | 1611 | 1.7 | |
| Missing | 1244 | 0.1 | 1045 | 1.0 | 472 | 0.0 | 575 | 0.6 | |
| Gestational age | | | | | | | | | |
| <28 | 6071 | 0.5 | 4661 | 0.4 | 4061 | 0.4 | 600 | 0.6 | <0.0001 |
| 28 | 2035 | 0.2 | 1524 | 0.1 | 1382 | 0.1 | 142 | 0.2 | |
| 29 | 2333 | 0.2 | 1739 | 0.2 | 1602 | 0.2 | 137 | 0.1 | |
| 30 | 2999 | 0.2 | 2261 | 0.2 | 2071 | 0.2 | 189 | 0.2 | |
| 31 | 3816 | 0.3 | 2791 | 0.2 | 2570 | 0.2 | 221 | 0.2 | |
| 32 | 5367 | 0.4 | 3820 | 0.3 | 3509 | 0.3 | 311 | 0.3 | |
| 33 | 7520 | 0.6 | 5581 | 0.5 | 5193 | 0.5 | 388 | 0.4 | |
| 34 | 12 261 | 1.0 | 9225 | 0.8 | 8506 | 0.8 | 719 | 0.7 | |
| 35 | 18 049 | 1.4 | 14 170 | 1.2 | 13 049 | 1.2 | 1122 | 1.2 | |
| 36 | 33 701 | 2.7 | 27 535 | 2.4 | 25 271 | 2.4 | 2264 | 2.3 | |
| 37 | 71 643 | 5.7 | 62 034 | 5.3 | 56 917 | 5.3 | 5117 | 5.2 | |
| 38 | 168 887 | 13.5 | 156 200 | 13.3 | 142 975 | 13.3 | 13 226 | 13.5 | |
| 39 | 272 469 | 21.8 | 260 443 | 22.3 | 238 741 | 22.3 | 21 707 | 22.2 | |
| 40 | 337 793 | 27.0 | 322 748 | 27.6 | 296 399 | 27.6 | 26 347 | 27.0 | |
| 41 | 242 264 | 19.4 | 234 547 | 20.0 | 215 217 | 20.1 | 19 325 | 19.8 | |
| 42 | 48 240 | 3.9 | 47 237 | 4.0 | 43 521 | 4.1 | 3716 | 3.8 | |
| 43+ | 5063 | 0.4 | 4893 | 0.4 | 4578 | 0.4 | 315 | 0.3 | |
| Missing | 9979 | 0.8 | 9381 | 0.8 | 7553 | 0.7 | 1829 | 1.9 | |

All live births = all live births within the Office for National Statistics (ONS) birth cohort.
All eligible births = all live, singleton, NHS hospital births to mothers living in England within the ONS birth cohort.
Linked = all eligible births from ONS birth cohort that successfully linked with an Hospital Episode Statistics (HES) birth record.
Unlinked = all eligible births from ONS birth cohort that did not successfully link with an HES birth record.
*$\chi^2$ test: linked versus unlinked.
IMD, Index of Multiple Deprivation.

Digital Economy Act 2017 is designed to enable research for public benefit through the sharing of data, but the legislation is limited to the sharing of data that has been deidentified. This means reliance on the Trusted Third Party model in research is likely to increase over time. It is still unclear how this affects health and social care data, especially as they are not covered by the Digital Economy Act.[4] If researchers do not have access to personal identifiers, QA of the linkage will become more challenging.

The findings in this paper will offer some insight into the quality of linkage between ONS birth and HES birth records and be of use to other researchers using the same linkage. As the quality and completeness of HES improved over time, it is likely that the quality of linkage will also have improved as a consequence, but the signs that the improvement in quality may not have been sustained is worrying.

### Recommendations for future practice and research

The findings from this study have demonstrated a need for Trusted Third Parties to evaluate linkage methods and publish findings for researchers to use prior to beginning statistical analysis. In cases where this is not possible, a number of studies have shown that datasets can be linked using deidentified datasets and produce generalisable cohorts; therefore, future work could explore ways to quality assure linkage without using personal identifiers. This will increase the likelihood of easier data access, potentially reducing the time required to complete the linkage and cleaning of routinely collected datasets. Finally, we did not assess whether the quality of linkage for multiples is comparable with live, singleton, NHS hospital births although QA of the linkage of multiple births to mothers' delivery records suggested that it was better than had been initially assumed.[6]

### CONCLUSIONS

By linking together administrative datasets, such as birth registration, birth notification and hospital admission data, it is possible to create more complete datasets about births which can then be linked to other administrative datasets to look at longer term outcomes.[6] If data are missing or there are coding errors and inconsistences, the resulting datasets can often be of poor quality. It is therefore essential that the quality of linkage is assessed prior to analysis. The work presented in this study provides a guide to steps taken to quality assure the linkage of births in England and may be of use to other researchers working with similar datasets. The findings will be particularly useful for researchers working with the same dataset but without access to personal identifiers.

**Acknowledgements** The authors are grateful to the following organisations and individuals who were involved in this study: the MRC for funding this project (MR/M01228X/1); the Office for National Statistics (ONS) for providing access to data and hosting it within the Secure Research Service (SRS); NHS Digital for linking and providing access to the data; Nirupa Dattani for providing her expertise on record linkage and data management; Rod Gibson, for his help cleaning the data; and Gillian Harper for her original work on the quality assurance of linkage of the mother's HES delivery records. This work contains statistical data from ONS, which is Crown Copyright. The use of the ONS statistical data in this work does not imply the endorsement of the ONS in relation to the interpretation or analysis of the statistical data. This work uses research datasets, which may not exactly reproduce National Statistics aggregates.

**Contributors** All authors were involved in the design of the study with input from Rod Gibson (Data management consultant) and Nirupa Dattani (Data analyst at City, University of London). RG and ND were involved in the cleaning of the ONS births dataset. VC performed the data manipulation and analysis. All authors were all involved in the interpretation of the data. VC was responsible for the initial draft of the manuscript. AM and MQ reviewed and contributed to drafts of the manuscript, and all authors have reviewed the final version.

**Funding** This work was funded by the Medical Research Council: MR/M01228X/1. VC and MQ had full access to all the data in the study and final responsibility for the decision to submit for publication.

**Disclaimer** The funder had no input into the study design, data analysis, interpretation of results or writing of the manuscript.

**Competing interests** None declared.

**Patient and public involvement** Patients and/or the public were involved in the design, or conduct, or reporting, or dissemination plans of this research. Refer to the Methods section for further details.

**Patient consent for publication** Not required.

**Ethics approval** Ethics approval for this study was granted by the Health Research Authority Research Ethics Committee (South West – Frenchay; REC reference *Ethics* 15/SW/0294). The TIGAR study used data linked as part of a previous study led by City, University of London.[6] For that study, ethics approval 05/Q0603/108 and subsequent substantial amendments were granted by East London and City Local Research Ethics Committee 1 and its successors. Permission to use patient-identifiable data without consent under Regulation 5 of the Health Service (Control of Patient Information) Regulations 2002 ('section 251 support') was initially granted by the Patient Information Advisory Group PIAG 2-10(g)/2005. Renewals and amendments and a second permission, CAG 9-08(b)2014, under Regulation 5 of the Health Service (Control of Patient Information) Regulations 2002 (or 'same legislation') were granted by the Secretary of State for Health and the Health Research Authority following advice from the Confidentiality Advisory Group (CAG) to use patient-identifiable data without consent and create a research database held at the ONS for analyses relating to inequalities in the outcome of pregnancy and to inform maternity service users about the outcome of midwifery, obstetric and neonatal care. For the TIGAR study, permission from the Health and Social Care Information Centre for the work described in this article was included in Data Sharing Agreement NIC-273840-N0N0 N.

**Provenance and peer review** Not commissioned; externally peer reviewed.

**Data availability statement** The authors do not have permission to supply data or identifiable information to third parties, including other researchers, but the team at City, University of London has permission under Regulation 5 of the Health Service (Control of Patient Information) Regulations 2002 to analyse patient-identifiable data for England and Wales without consent and create a research database that could be accessed by other researchers using the SRS at the ONS. The TIGAR team has permission under Regulation 5 of the Health Service (Control of Patient Information) Regulations 2002 to analyse these. Anyone wishing to access the linked datasets for research purposes should apply via the Confidentiality Advisory Group (CAG) to the Health Research Authority to access patient-identifiable data without consent and then to the ONS and NHS Digital. In the first instance, enquiries about access to the data should be addressed to Alison Macfarlane.

**ORCID iD**
Victoria Coathup http://orcid.org/0000-0003-0557-6757

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
