## [Reviewer comments · BMJ Open]

ARTICLE DETAILS

TITLE (PROVISIONAL)	Linkage of Maternity Hospital Episode Statistics birth records to birth registration and notification records for births in England 2005-2006: Quality assurance of linkage
AUTHORS	Coathup, Victoria; Macfarlane, Alison; Quigley, Maria

VERSION 1 – REVIEW

REVIEWER	Rachael Wood NHS National Services Scotland Information Services Division Scotland UK
REVIEW RETURNED	13-Mar-2020

GENERAL COMMENTS	This study provides information on the quality assurance of record linkage between ONS birth registration/notification data and HES maternity (birth) admission record data. Singleton births in 2005 and 2006 to mothers living in England are covered. The data sets involved are well described and data cleaning and quality assurance methods are clearly presented. Results are broadly reassuring. 92% of ONS birth records were successfully linked to a HES maternity (birth) record. Incorrect links were uncommon. Systematic differences between the characteristics of linked and unlinked births (and hence potential for bias in analysis of the linked dataset) were trivial. My only specific request for clarification is as follows. The final para p6 / first para p7 describing mother and baby records within the HES maternity dataset is difficult to follow – for example how are the 19 'baby tail' variables distributed across the maternal and baby records? Broader comments are as follows. The CHI squared results from comparison of characteristics of linked and unlinked records are not incorrect per se however arguably they are not helpful due to the very high sample sizes involved. Results tend to be 'significant' even when differences are trivial. The authors are right to focus on absolute differences of >1% in the discussion text. The births covered by this study occurred in 2005 and 2006. Very high missing data rates for key variables such as birthweight and gestation are evident in the HES maternity data. Additional specific information on subsequent trends in data completeness rates would help readers to assess how applicable these results may be to more recent data. This study highlights a key tension in research based on linkage of routine records: how to balance privacy enhancing approaches (such as use of trusted third parties) and transparency/ability to assess the quality of linkage and hence quantify any bias introduced
--

	through the linkage process. Linkage organisations and research teams need to work together provide relevant information, for example as set out in the GUILD framework.
--	--

REVIEWER	Mati Rahu National Institute for Health Development, Estonia
REVIEW RETURNED	17-Mar-2020

GENERAL COMMENTS	It is a valuable contribution to a generally neglected subject. I have only some minor comments that could probably help to improve the clarity of the paper:  1. The title of the paper is too long and complex. Avoid repeating the words „linkage“ and „Hospital Episode Statistics“. 2. Define the acronym „NHS“. 3. No need to create the acronym „DEA“ because it has not been used in the following text. 4. Table 2. Delete vertical lines. 5. Tables 3 and S1. Replace total % 100.0 with 100. 6. Table 4. Create the title for the 1st column (e.g. Characteristic). Unreadable text under the table. 7. Table S4. Delete the percentage sign (%) from the table head.
--

REVIEWER	Ann Sprague BORN Ontario Ottawa, ON Canada
REVIEW RETURNED	18-Mar-2020

GENERAL COMMENTS	Thanks for asking me to review this paper. It is great to review a well-written paper. The topic is important as data quality is key to consider when using administrative decisions to make health policy or health funding decisions. I have concerns because this data was collected 14-15 yrs ago and would like to know if the data quality might have improved significantly since then. Did you consider comparing it to more recent data, perhaps 10 yrs later (perhaps 2015-2016)? I think it is essential to add a paragraph to the introduction explaining how data is entered in each of these systems you discuss. Is it manually done (and by whom) or is it extracted from a health records system? Also what is the time lag from when the birth event happens and when the data is entered This will help the reader understand the context of any data quality issues. Is there special training of any sort for the people doing the data entry? How long does it take to enter the record? In regards to methods, I wonder why you didn't use a weighting system when trying to resolve the linkages. That is pretty standard practice for probabilistic and deterministic linkage work. Some variables are weighted higher than others and then you establish cut-offs for a successful link. My other editing comments are contained in notes in the attached manuscript – please contact publisher.
---

VERSION 1 – AUTHOR RESPONSE

Reviewer 1

1. This study provides information on the quality assurance of record linkage between ONS birth registration/notification data and HES maternity (birth) admission record data. Singleton births in 2005 and 2006 to mothers living in England are covered.

2. The data sets involved are well described and data cleaning and quality assurance methods are clearly presented.

3. Results are broadly reassuring. 92% of ONS birth records were successfully linked to a HES maternity (birth) record. Incorrect links were uncommon. Systematic differences between the characteristics of linked and unlinked births (and hence potential for bias in analysis of the linked dataset) were trivial.

4. My only specific request for clarification is as follows. The final para p6 / first para p7 describing mother and baby records within the HES maternity dataset is difficult to follow – for example how are the 19 'baby tail' variables distributed across the maternal and baby records?

Thank you for highlighting this. We have now added further detail explaining what is included in each type of HES Maternity record. Please see page 5 paragraph 1 and page 6, paragraph 4.

5. Broader comments are as follows. The CHI squared results from comparison of characteristics of linked and unlinked records are not incorrect per se however arguably they are not helpful due to the very high sample sizes involved. Results tend to be 'significant' even when differences are trivial. The authors are right to focus on absolute differences of >1% in the discussion text.

6. The births covered by this study occurred in 2005 and 2006. Very high missing data rates for key variables such as birthweight and gestation are evident in the HES maternity data. Additional specific information on subsequent trends in data completeness rates would help readers to assess how applicable these results may be to more recent data.

We have added more detail into the discussion on this and hope it is now clearer for the reader. Please see page 17, paragraph 4 (interpretation of findings). We have also referenced the previous study's final report in which there is detailed information.

7. This study highlights a key tension in research based on linkage of routine records: how to balance privacy enhancing approaches (such as use of trusted third parties) and transparency/ability to assess the quality of linkage and hence quantify any bias introduced through the linkage process. Linkage organisations and research teams need to work together provide relevant information, for example as set out in the GUILD framework.

Reviewer 2

8. The title of the paper is too long and complex. Avoid repeating the words „linkage“ and „Hospital Episode Statistics“.

The title has now been edited and is highlighted in yellow.

9. No need to create the acronym „DEA“ because it has not been used in the following text

This has been amended in the introduction.

10. Define the acronym „NHS“.

This has been defined in the introduction.

11. Table 2. Delete vertical lines.

These have been removed from the table. Please see Table 2 on page 12.

12. Tables 3 and S1. Replace total % 100.0 with 100.

These have been amended. Please see Table 3 on page 13 of the main manuscript and Table S1 on page 3 of the supplementary information.

13. Table 4. Create the title for the 1st column (e.g. Characteristic). Unreadable text under the table

Characteristic has been added to column 1 and the text below the table has been increased in size. We hope this is now readable. Please see Table 4 on pages 15 and 16 (Highlighted in yellow).

14. Table S4. Delete the percentage sign (%) from the table head

These have been removed from the table. Please see Table S4 on page 6 of the supplementary information.

Reviewer 3

15. It is great to review a well-written paper. The topic is important as data quality is key to consider when using administrative decisions to make health policy or health funding decisions.

16. I have concerns because this data was collected 14-15 yrs ago and would like to know if the data quality might have improved significantly since then. Did you consider comparing it to more recent data, perhaps 10 yrs later (perhaps 2015-2016)?

The linkage and quality assurance on linkage was performed as part of a larger study (the TIGAR study), which explored gestational age and long term health outcomes. Therefore, we had to ensure we had adequate follow up to look at outcomes up to 10 years of age. Analysing subsequent years of data solely for the purpose of comparing linkage was beyond the remit of this study, but there is further information in the final report of the previous study (Macfarlane A, Dattani N, Gibson R, Harper G, Martin P, Scanlon M, et al. Births and their outcomes by time, day and year: a retrospective birth cohort data linkage study. *Heal Serv Deliv Res.* 2019;7(18):1–268.). However, we do acknowledge that this would have been interesting and useful for other researchers in this area, and therefore have added additional detail on later rates in the discussion. Please see page 17, paragraph 4 (Interpretation of findings).

17. I think it is essential to add a paragraph to the introduction explaining how data is entered in each of these systems you discuss. Is it manually done (and by whom) or is it extracted from a health records system? Also what is the time lag from when the birth event happens and when the data is entered. This will help the reader understand the context of any data quality issues. Is there special training of any sort for the people doing the data entry? How long does it take to enter the record?

Additional information about this has been added at the bottom of page 4 and top of page 5.

18. In regards to methods, I wonder why you didn't use a weighting system when trying to resolve the linkages. That is pretty standard practice for probabilistic and deterministic linkage work. Some variables are weighted higher than others and then you establish cut-offs for a successful link

A weighting system was considered when assessing the quality of the linkage. However, our work

built on previous work by City University, who quality assured the linkage between birth registration, birth notification records and Maternity HES Delivery records. The algorithms designed for that study had proven to be effective, so we built on this and adapted the algorithm to suit the HES Birth records. The algorithm designed is simple and easy to use by other researchers who wish to assess linkage to later years.

19. Very long title, could probably cut it after Quality assurance of linkage.

This has been amended as suggested and is highlighted in yellow.

20. What happens for home births - is it the same? What proportion of home births happen in UK

Records are created in the same way for home births, but many home births are not captured on hospital systems. We have added additional information about this on page 5, paragraph 1 of methods section.

21. Would be nice to have one sentence on the results of that study

We have added some detail about the findings and we also describe the findings within the discussion. Please see the final paragraph of the introduction.

22. You just said in the sentence above that you have already done this?

We have now edited this sentence.

23. why not multiples?

Whilst we acknowledge that it would have been useful for other researchers if we had published findings on multiple births, the quality assurance of the linkage was performed as part of the TIGAR study, which was exploring the effect of gestational age on long term outcomes in children in singleton births. Therefore, multiple births were excluded.

24. I have to say that reading this (not being from the UK health care system) is a bit confusing. I'm constantly having to go back and figure out what each system acronym refers to. Is there a way to simplify this for other readers: perhaps it could be the hospital birth records, the national statistics birth record, and the national ID # or whatever makes sense.

We agree that the terms can be slightly difficult to follow, however, we believe it is important to remain transparent about the data sources used in this study. The UK is four nations, each with its own system of hospital statistics and a new system is being developed for maternity statistics for England. Therefore, we have kept the terms as they were, but have added in additional information that we hope has clarified things for the reader. Please see pages 4, 5 and 6 where we have included further descriptions of the data sources used.

25. should this be births in England?

This has been amended.

26. what does this mean?

Further description of the HESID has been added to paragraph 2 on page 6.

27. is the example meant to make the point about episodes of care that happen between two fiscal year's? If so, should probably say

This point has been made clearer in the methods section. Please see page 6, paragraph 3.

28. spell isn't a particularly universally understood term. Is there a better word?

While we agree this is a slightly confusing term, we wish to remain consistent with other published literature in this area. However, we have added additional detail into the data sources section and defined spell and episode for readers. We hope this is now easier to follow. Please see section B. Hospital Episode Statistics (HES Birth records) on page 6.

29. I'm wondering why you didn't weight the variables as to importance for the quality of the linkage. Your table 1 provides the different options, but presumably some of the variables for linkage would be weighted higher than others and thus lead to higher confidence? That seems to be a common methods associated with deterministic and probabilistic linkages - any reason you didn't do this?

Please see response to comment 18 above.

30. I think it says one less in the table: 1,074,571

Numbers in tables, text and figures have all been checked and corrected where there are small errors.

31. Says one more in the table 96, 219

Numbers in tables, text and figures have all been checked and corrected where there are small errors.

32. Just for the discussion - make sure you discuss how this data (Manually or extracted from an electronic health record system) is entered and if that makes a difference in missingness.

This is now been explained more fully on page 5, paragraph 1.

33. probably should change this to multiple care episodes since multiple birth tends to imply a different concept

Thank you for highlighting this. We have now changed this.

34. There seems to be a problem with how the numbers rendered in these columns as they don't add up to the total. Looks like there are some digits missing on some

Some of the columns were not wide enough and had cut off the final numbers. Therefore, we have now amended the column sizes and hope this is clearer. Please see Table 4 on pages 15 and 16.

35. Wouldn't it be important for the trusted 3rd parties to evaluate and publish their QA in order to be awarded contracts for the work?

We agree, but the Trusted Third Party was the data owner and we had no option of choosing a different one. At the time when earlier work was being done, one of us searched extensively for an evaluation of its quality assurance procedures, but could not find one. This was mentioned in the following report and paper:

1. Macfarlane A, Dattani N, Gibson R, Harper G, Martin P, Scanlon M, et al. Births and their outcomes by time, day and year: a retrospective birth cohort data linkage study. *Heal Serv Deliv Res.* 2019;7(18):1–268.
2. Harper G. Linkage of Maternity Hospital Episode Statistics data to birth registration and notification records for births in England 2005-2014: Quality assurance of linkage of routine data for singleton and multiple births. *BMJ Open.* 2018;8(3).

We have also explored this further within the discussion. Please see page 18, paragraphs 3 and 5.

36. or other administrative datasets to look at longer term outcomes

This has been amended in the text. Please see page 19, paragraph 1.

VERSION 2 – REVIEW

REVIEWER	Ann Sprague Better Outcomes Registry and Network (BORN) Ontario Canada
REVIEW RETURNED	23-Jun-2020
GENERAL COMMENTS	Thanks for the work in responding and editing the manuscript. I am satisfied with the revision completed. I made a couple of minor editorial suggestions and one comment that I'm sure would be picked up in the final editing process.

VERSION 2 – AUTHOR RESPONSE

Reviewer 3

Minor comments:

1. Amend sentence in abstract

This has now been amended.

2. Highlight Figure S1 in introduction

We have referred the reader to Figure S1 in the introduction section now.

3. Rephrase sentence on page 6

This has been amended as suggested.

4. Amend sentence on page 15 – incorrect tense

This has been amended.

5. Why were numbers in table changed?

These were highlighted in the previous reviewer comments and so we double-checked the regional

figures and some did not report the most up to date figures. They were therefore amended and all other figures reported were also checked, but no other errors were found.